# The Burden of Out-of-Pocket Expenditure Related to Gynaecological Cancer in Malaysia

**DOI:** 10.3390/healthcare10102099

**Published:** 2022-10-20

**Authors:** Chee Hui Liew, Fatiha Hana Shabaruddin, Maznah Dahlui

**Affiliations:** 1Department of Social and Preventive Medicine, Faculty of Medicine, Universiti Malaya, Kuala Lumpur 50603, Malaysia; 2Department of Pharmacy, Hospital Kuala Lumpur, Jalan Pahang, Kuala Lumpur 50586, Malaysia; 3Department of Clinical Pharmacy and Pharmacy Practice, Faculty of Pharmacy, Universiti Malaya, Kuala Lumpur 50603, Malaysia; 4Centre of Population Health, Department of Social and Preventive Medicine, Faculty of Medicine, Universiti Malaya, Kuala Lumpur 50603, Malaysia

**Keywords:** out-of-pocket expenditure, gynaecological cancer, catastrophic health expenditure, financial burden, impoverished

## Abstract

This study aimed to estimate the economic burden on gynaecological cancer patients and their households, in terms of out-of-pocket expenditure, catastrophic health expenditure (CHE) and poverty impact. A cross-sectional study was conducted at an academic tertiary-care health centre in an upper-middle-income country. Data were obtained via structured interviews of 120 gynaecological cancer patients alongside review of medical charts. Mean (SD) and median (IQR) annual household out-of-pocket expenditures were USD 2780 (SD = USD 3926) and USD 1396 (IQR = 3013), respectively. Two thirds (*n* = 77/120, 64%) of households experienced CHE and 17% (*n* = 20/120) were impoverished due to out-of-pocket expenditure related to gynaecological cancer. Factors associated with CHE, explored using multivariate logistic regression analysis, estimated that the highest income quintile households, Q5, were 90% less likely to incur CHE compared to the lowest income quintile households, Q1 (adjusted odds ratio = 0.100; *p*-value < 0.05) and that patients who were not receiving chemotherapy were 88% less likely to incur CHE compared to those receiving chemotherapy (adjusted odds ratio = 0.120; *p*-value < 0.05). These results indicate the necessity to broaden the coverage of existing financial assistance for patients from low- and middle-income households, such as extending coverage to adult patients of all ages and to those treated in all public hospitals, including academic health centres.

## 1. Introduction

The global cancer burden is rising and is expected to increase further in the coming decades [1]. In female patients, gynaecological cancers are among the top cancers in Malaysia with high mortality [2,3,4]. Ovarian, cervical, and endometrial cancer are the three most common gynaecological cancers worldwide as well as in Malaysia [3,5,6]. The increasing clinical burden of gynaecological cancer is costly, in terms of direct and indirect costs, and from both healthcare and societal perspectives [7,8]. Treatment for patients with cancer generally involves high-cost clinical strategies and medications. Gynaecological cancer treatment can cause a significant financial burden on both families and society [9]. Currently, there are scarce local data on the economic impact of cancer, including the healthcare expenditure of patients with gynaecological cancers in Malaysia.

The total economic burden of a disease comprises direct medical and non-medical costs, as well as indirect costs such as productivity losses. A diagnosis of cancer often leads to substantial economic implications on patients and their households [9,10,11]. The financial implications due to high out-of-pocket healthcare-related expenditure can negatively affect the lives of patients and their households [9,12]. In Malaysia, there is healthcare-related financial assistance available for citizens from low- and middle-income households, on top of the availability of partially subsidised healthcare for all citizens in government-run healthcare facilities nationally. National policies in Malaysia often target specific income groups, defined based on households’ monthly income, comprising the low-income B40 households (monthly household income of <RM 3000 (<USD 683)), middle-income M40 households (monthly household income of RM 3000–13,148 (USD 683–2992)) and high-income T20 households (monthly household income of >RM 13,148 (>USD 2992)) [13]. The two national healthcare-related financial assistance schemes are the MySalam and the Peka B40 schemes, which were implemented by the government within these past 5 years to assist patients from B40 and M40 households [14,15]. The MySalam scheme offers one-off financial assistance to adult patients up to 65 years old from low-(RM 8000 (USD 1820)) and middle-income (RM 4000 (USD 910)) households who have been diagnosed with any of the diseases listed as a critical illness, which include cancer. The MySalam scheme also provides recipients with hospitalisation allowances for up to 14 days per annum for admissions in government-run health facilities [15]. In addition to MySalam, the Peka B40 scheme also provides financial assistance for patients from low-income B40 households with their health-related expenditures when they receive treatment in public hospitals run by the Ministry of Health [14]. The Peka B40 financial assistance includes medical equipment assistance, cancer treatment incentives and transport incentives. While these national schemes may alleviate the economic burden of cancer patients, cancer is often an expensive diagnosis and many patients incur substantial out-of-pocket expenditure for a prolonged period after diagnosis. Gynaecological cancer guidelines recommend multiple treatment modalities for gynaecological cancers, including chemotherapy, radiotherapy, surgery and hormonal therapy [16,17,18,19,20], leading to lengthy durations of treatment.

Out-of-pocket expenditure (OOP) is defined as financial expenditures of hospital and non-hospital health-related costs that were directly paid by patients and their households, including direct non-medical costs such as patients’ transportation and medical fees [21]. OOP expenditure can be used to quantify the direct financial impact of an illness on patients and their households. Catastrophic health expenditure (CHE) can be defined as OOP expenditure that exceeds a certain proportion of a household’s income in a given time period, usually one year [22]. If the healthcare expenditure is large compared to the financial resources available to the household, the potential disruption to the living standards may be considered catastrophic [22]. Various thresholds can be used to define catastrophic expenditure, such as spending 10%, 20%, 30% or 40% of the total household income on health-related expenses [23].

There are many factors that could influence OOP and CHE. One study found that patients in Malaysia who were diagnosed with advanced stages of colorectal cancer (stage III and IV) incurred higher OOP expenditure compared to those diagnosed with earlier stages of cancer [11]. Another study investigating OOP expenditure among breast cancer patients in Malaysia found that the highest proportion of OOP spending was on adjuvant therapy, followed by complementary and alternative therapy [24]. A study on OOP expenditure of cancer patients in Canada found travel cost to be higher compared to other costs such as accommodation and supplements [25]. Some of the factors that have been found to be associated with CHE included households with low income, patients receiving chemotherapy, patients without insurance, households with members who were hospitalised, elderly, or chronically ill, and households in rural or poorer regions [8,11,26,27].

This study aimed to describe the economic burden of gynaecological cancer on patients and their households, in terms of assessing OOP expenditure and CHE among patients with the three most common gynaecological cancers (ovarian, cervical and endometrial cancers) who were treated at an academic tertiary-care healthcare centre in Malaysia, an upper-middle-income country in Asia. This study also aimed to identify factors associated with CHE among the study population. The poverty impact of gynaecological cancer patients was also explored as the secondary objective.

## 2. Materials and Methods

This cross-sectional study was approved by the UMMC Medical Research Ethics Committee (MREC), with MREC number 201936-7203. The inclusion criteria were: patients with ovarian, cervical or endometrial cancer, adult patients (≥18 years old), patients who were diagnosed with either ovarian, cervical or endometrial cancer and were treated for at least 12 months at the study institution. The exclusion criteria were: patients with multiple cancer diagnoses and patients of other nationalities, as they are not expected to reflect the OOP expenditure and CHE of the local gynaecological cancer population due to their small number as well as differences in clinical needs and access to subsidised healthcare and financial assistance.

### 2.1. Data Collection

Data collection involved structured face-to-face interviews using a structured study questionnaire and was supplemented by review of medical charts for clinical data including data on clinical treatment received in the last 12 months. The recall period for health-related OOP expenditure was for the past 12 months before the date of the interview. The study questionnaire was developed based on published literature [8,11,25,26,28] and was available in English and Malay, the national language. Written informed consent was obtained from all participants prior to the interviews. There were no tokens given for participation. Data collection was conducted for 6 months between July until December 2019. We attempted to invite all ovarian, cervical and endometrial cancer patients who were receiving treatment at the study institution during the data collection period and who met the inclusion criteria to participate in this study. Patients who had received treatment and been followed up at the clinic for at least 12 months were identified from the daily clinic list and were invited to participate. There were 120 patients who agreed to participate in this study out of the 128 patients who were invited, leading to a 93.8% (*n* = 120/128) participation rate. Of the eight patients who declined to participate, four patients cited time constraints, while another four were unwilling to disclose their financial details.

Self-reported data on patients’ OOP expenditure comprised all health-related expenditures including medical and hospital fees as well as non-medical expenditures directly paid by patients that were not reimbursed by employers or other agencies. Direct non-medical expenditures included payments for transportation, car parking charges, accommodation and food during clinic visits or hospital admission. In addition, payments for nursing care or nursing home, complementary and alternative therapy, over-the-counter drugs, vitamins and supplements, medical devices or equipment and homecare expenses were also included. The medical and hospital-related payments, such as for medications, laboratory investigations, diagnostic imaging tests, consultation fees and any other medical charges for any health treatments were also included.

In this study, CHE was defined as annual OOP expenditure exceeding 10% of annual household income, similar to several published studies [8,22,29]. For annual income, patients were asked to report their monthly household income during the interview, which was multiplied by 12 to estimate total household income per annum. Households were considered poor based on the national poverty threshold, defined as monthly household income below RM 2208 (USD 502) in Malaysia [30]. Impoverishment due to health-related OOP expenditure was defined as expenditure that results in a household falling below the poverty line, where if the health-related expenditure was not considered, the household would have been above the poverty line [22].

### 2.2. Statistical Analysis

Data were analysed with SPSS version 24.0 (SPSS Inc., Chicago, IL, USA). The association between all of the independent variables in Table 1 and dependent variables were assessed using binary logistic regression analysis. Following the bivariate analysis, univariate logistic regression analysis was conducted individually for variables with significant results (*p*-value of <0.05) in the bivariate analysis. Variables that were significantly associated with CHE in the univariate logistic regression models (*p*-value of <0.25) were included in the final multivariate logistic regression model. The final results were interpreted as significantly associated with CHE using the *p*-value of < 0.05. The goodness-of-fit of the logistic regression model was checked by applying the Nagelkerke R square and Hosmer–Lemeshow test.

Costs are presented in Malaysian Ringgits (RM) and in United States Dollars (USD) for international comparison, at a conversion rate of USD 1 = RM 4.3950 (Bank Negara Malaysia Exchange Rates on 22 April 2020) [31]. The results on OOP expenditure, CHE and impoverishment were presented according to income quintile groups to facilitate comparisons with international published studies. In addition to quintiles, results were also presented according to patients’ household income categories that are relevant to national policies in Malaysia.

## 3. Results

Table 1 summarises the sociodemographic and clinical characteristics of the 120 study participants. There were 85 (70.8%) patients who had at least secondary level education. Mean and median distance from patients’ residences to the healthcare centre was 106 km (SD = 253) and 26 km (IQR = 56), respectively. Two of the nine patients with some form of health insurance received critical illness compensation from their insurance providers. Most patients (*n* = 96/120, 80%) were retired or were currently not in the workforce. Mean and median household incomes for the 120 households were RM 5031 (SD = RM 7136) (USD 1145 (SD = USD 1624)) and RM 3000 (IQR = 4300) (USD 683 (IQR = 978)), respectively. Mean and median household sizes were 4 (SD = 2) and 3 (IQR = 3), respectively. Table 1 also presents the clinical characteristics of the study population. Patients’ stage of cancer was based on their latest cancer staging on the day of their interviews. Of the 10 patients (8.3%) who experienced disease progression during the 12 months’ recall period, eight were diagnosed 2 to 5 years ago, one patient was diagnosed around 1 year ago and another was diagnosed more than 11 years ago. Only 45 (37.5%) patients were receiving active cancer treatment (patients on cancer therapy) during the 12 months’ recall period, while 75 (62.5%) patients were attending the oncology clinic for scheduled follow-ups and clinical surveillance.

### 3.1. Health-Related Out-of-Pocket Expenditure

The mean (SD) and median (IQR) annual health-related OOP expenditure for households with gynaecological cancer (*n* = 120) was RM 12,218 (SD = RM 17,255) (USD 2780 (SD = USD 3926)) and RM 6134 (IQR = RM 13,243) (USD 1396 (IQR = USD 3013)), respectively. The results of the household OOP expenditure by category, including percentage, mean, standard deviation, median and interquartile range are shown in Table 2.

Health-related OOP expenditure and the proportion of households experiencing CHE and living below the poverty line are presented according to income quintile groups in Table 3. In addition to quintiles, patients’ households were also divided into income groups that were relevant to national policies in Malaysia and the results are reported according to these income categories in Table 4. Generally, households with higher income had higher OOP expenditure compared to households with lower income.

### 3.2. Poverty Impact

Based on monthly household income alone before considering any health expenditure, there were 45 households (37.5%) with monthly household incomes below the national poverty line of RM 2208 (USD 502) (30). There were 20 gynaecological cancer patients’ households that were impoverished due to OOP expenditure incurred in the past 12 months, which led to a total of 65 households that lived below the national poverty threshold.

### 3.3. Catastrophic Health Expenditure

Two thirds of the study households (77/120 (64.2%)) experienced CHE. The variables that were significantly associated with CHE in bivariate analysis were household income, cancer stage and receiving chemotherapy. The association between variables and CHE were then explored using univariate and multivariate regression analysis, and the results are presented in Table 5 and Table 6. The Nagelkerke R^2^ and Hosmer–Lemeshow tests showed satisfactory goodness-of-fit for the regression models. The model explained 30.1% (Nagelkerke R^2^) of the variance in CHE and correctly classified 72.5% of cases. The Hosmer–Lemeshow test showed satisfactory goodness-of-fit for the regression model (*p*-value = 0.734). Households were categorised according to income quintiles in the analysis presented in Table 5, which estimated that the highest income quintile households, Q5 (adjusted odds ratio = 0.100; 95% CI = 0.025–0.393; *p*-value < 0.05), were 90% less likely to incur CHE compared to the lowest income quintile households, Q1. Patients who were not receiving chemotherapy were 88% less likely to incur CHE compared to those receiving chemotherapy (adjusted odds ratio = 0.120; 95% CI = 0.033–0.443; *p*-value < 0.05).

Households were also divided according to income categories that are relevant to national policies. The results of the analysis are presented in Table 6, and show that middle-income M40 households (adjusted odds ratio = 0.225; 95% CI = 0.090–0.567; *p*-value < 0.05) and higher income T20 households (adjusted odds ratio = 0.048; 95% CI = 0.005–0.500; *p*-value < 0.05) were 77% and 95% less likely to incur CHE respectively compared to low-income B40 households. This analysis also found that patients who were not receiving chemotherapy were 84% less likely to incur CHE compared to those receiving chemotherapy (adjusted odds ratio = 0.159; 95% CI = 0.049–0.523; *p*-value < 0.05). No other variables were found to be significantly associated with CHE in the final multivariate regression analyses.

**Table 6 healthcare-10-02099-t006:** Estimated association between variables and catastrophic health expenditure using univariate and multivariate regression analysis according to household income categories (*N* = 120).

Variables	Univariate Crude Odds Ratio (95% Confidence Interval)	*p*-Value	Multivariate Adjusted Odds Ratio (95% Confidence Interval)	*p*-Value
Household income categories				
Low-income B40	Reference		Reference	
Middle-income M40	0.399 (0.178–0.896)	0.026	0.225 (0.090–0.567)	0.002 *
High-income T20	0.060 (0.006–0.565)	0.014	0.048 (0.005–0.500)	0.011 *
Stage of cancer at the beginning of the 12 months’ recall period				
Stage I and stage II	Reference		Reference	
Stage III and stage IV	2.762 (1.264–6.033)	0.011	2.026 (0.821–5.001)	0.126
Cancer treatment received within 12 months’ recall period				
Chemotherapy				
Yes	Reference		Reference	
No	0.206 (0.073–0.583)	0.003	0.159 (0.049–0.523)	0.002 *

* statistically significant at *p* < 0.05.

## 4. Discussion

There is a scarcity of published studies on OOP expenditure and CHE of gynaecological cancer patients globally and especially within the region. Existing studies indicated that gynaecological cancer patients’ households were highly affected by financial difficulties, such as in Thailand [32] and the United States [33]. The mean annual OOP expenditure of gynaecological cancer patients’ households in this study was RM 12,218 (SD = RM 17,255) (USD 2780, SD = USD 3926). The components of OOP expenditure in this population that contributed to high annual expenditures were medical charges and fees, complementary and alternative therapy, vitamins and supplements, and transportation, and components with relatively lower expenditures were nursing home, medical device and accommodation expenses. This reflects that the study population spent comparatively more on complementary and alternative therapy and also vitamins and supplements in addition to hospital medical charges and transportation when compared to a study conducted in Canada [25]. This high use of complementary and alternative medicine is in line with finding from other local studies indicating that patients in Malaysia tend to use a combination of complementary and alternative medicine alongside conventional medications [24,34]. These alternative therapies are yet to be proven to have clinical effectiveness in cancer populations [35], which emphasises the need for health education on appropriate and rational use of medicine to limit excessive health-related expenditure and reduce the financial burden on patients.

While the quantitative OOP expenditure value in this study population is relatively low compared to international currencies, two thirds of the study households (64%) experienced CHE. This indicates that the impact of OOP expenditure was still considerable despite the availability of partially-subsidised healthcare for all citizens in government-run healthcare facilities in Malaysia, with varying levels of subsidy available in public healthcare centres nationally.

The annual mean OOP expenditure in this study was lower than found in a similar patient population in the United States [33] (RM 27,658 (USD 6293)), which is expected, as OOP expenditure is known to vary between countries due to differences in healthcare systems and the extent of financial health assistance and subsidised medical coverage [8,33]. Compared to the OOP expenditure of another cancer population in Malaysia, the annual OOP expenditure of gynaecological cancer patients in this study was higher than the OOP expenditure reported by colorectal cancer patients in Malaysia [11] (RM 8307 (USD 1890)). This implies that gynaecological cancer patients may have a higher financial burden than other cancer patients. Generally, ovarian and cervical cancers have low remission rates [36], and that may increase the clinical need for additional treatment strategies and multiple lines of treatment, which may lead to higher financial expenditures and economic burden.

A study on cancer patients in the ASEAN region reported that 48% of the study households experienced CHE, a slightly lower proportion than what was observed in this study. The ASEAN study was based on pooled results of several countries involving patients with various types of cancer [8]. The study of colorectal cancer patients in Malaysia found that nearly half (47.8%) of the patients experienced CHE [11]. A study in another Asian country, Iran, found that 68% of cancer patients’ households experienced CHE (26). The high proportion of CHE in those studies as well as in this study indicates that the economic impact of OOP expenditure on cancer patients and their households was significant and is expected to rise alongside the clinical burden of cancers in the coming decades.

There were two key variables found to be significantly associated with CHE in this study. First, households with lower incomes were significantly more likely to experience CHE compared to higher-income households. The significantly lower odds of middle- and high-income households experiencing CHE compared to low-income households were consistent with other studies [8,11,26,33,37]. Second, patients who received chemotherapy were significantly more likely to experience CHE. It is likely that the increased financial burden of additional hospital visits for cyclical and often lengthy chemotherapy regimens and related treatment resulted in higher medical fees and health-related expenditures [38]. The study on colorectal cancer patients in Malaysia also reported similar findings, where patients treated with chemotherapy had higher odds of experiencing CHE [11].

Despite the higher mean expenditure of patients with advanced cancer, this study did not find a statistically significant difference between patients with early and advanced cancer stages, similar to a study in the United States that reported higher healthcare expenditure in gynaecological cancer patients at advanced stages [33]. This still indicates the need for increased health awareness and health campaigns to encourage regular medical screening for early diagnosis of cancer. Moreover, patients in advanced stages are more likely to undergo chemotherapy and thus are at high risk of CHE. Early diagnosis can optimise clinical outcomes and healthcare resource use as well as reduce patients’ financial burden [8].

The eligibility for the main healthcare-related financial assistance for low- and middle-income households, MySalam, is currently limited to adult patients up to 65 years old [15]. In this study, nearly half of the patients were above 60 years old, indicating a need to broaden the coverage of the national MySalam scheme to include patients of all ages, particularly geriatric patients who are expected to increase in number in the coming decades. A study in China found that implementation of new medical schemes can reduce the financial burden of the elderly [39]. A study on CHE among the elderly in Malaysia found that among the comorbidities that were investigated, cancer was the only comorbidity that influenced CHE among the elderly, further highlighting the need to alleviate the financial burden on cancer patients nationally, especially among the elderly [40]. The national MySalam scheme is relatively new and was first implemented in January 2019. It was expected to directly benefit nearly 8 million individuals from the low-income B40 group who are in need of healthcare protection, representing 19% of the Malaysian adult population. Benefits of the MySalam scheme have been extended and broadened twice since its implementation, in early 2020 and again in 2022, indicating a recognition of the increasing healthcare needs of Malaysians [41]. Historically, Malaysia allocates a relatively low allocation for healthcare based on the country’s gross domestic product (GDP), generally less than 3% of national GDP. The Ministry of Health recently highlighted that past allocation of 2.59% of GDP for public health expenditure [42] was insufficient for the needs of the country and is requesting an increase, with the benchmark of 4–5% of GDP similar to other upper-middle-income countries. It is hoped that a future increase in the country’s healthcare budget would include prioritising extending financial assistance coverage to all adult Malaysians without any age limit.

The Peka B40 scheme is available for patients from low-income households who are 40 years old and older but is presently limited only to those receiving treatment in public hospitals run by the Ministry of Health. The Peka B40 financial assistance, which includes cancer treatment incentives and transport incentives, is not available to patients who are treated in academic tertiary-care hospitals, such as the study institution [14]. A previous study in Malaysia found that patients treated in a university hospital had higher OOP expenditure compared to those treated in a Ministry of Health public hospital [43]. It was beyond the scope of this study to explore why patients chose to seek treatment in this academic tertiary-care hospital rather than a public hospital run by Ministry of Health, despite the higher medical charges due to the lower rates of subsidy in the study institution. It is possible that patients’ preference could be based on proximity to their homes, perceived shorter waiting time or higher confidence in the clinical staff and the healthcare institution. The high proportion of households that experienced CHE in this study indicates the need to extend the coverage of the Peka B40 scheme to more public hospitals, including university hospitals, as well as potentially extending the scheme to patients from middle-income households. These financial challenges faced by patients due to the varying levels of financial assistance and healthcare subsidies in different healthcare institutions in Malaysia may be similar to healthcare settings in other middle-income countries.

Findings from this study can be used to tailor financial assistance, subsidy schemes and support to patient groups that are the most in need. Prioritising specific socioeconomic groups in equitable health policies for affordable cancer care can ensure satisfactory social and financial protection for patients from the most vulnerable groups. Future strategies can include reimbursement of OOP expenditures for cancer treatment, incentivising public sector and industry partnerships to develop mutually beneficial equitable patient access schemes, increasing the rates and coverage of healthcare subsidies for low- and middle-income households and extending the age limit of the recipients of healthcare-related financial assistance. Increased financial subsidy and assistance for medical charges and transportation to healthcare centres could also be helpful for patients from low- and middle-income households who are receiving chemotherapy.

There were several challenges faced when conducting this study. The relatively small number of patients compared to studies from other countries reflected the best of our attempts to invite nearly all of the eligible gynaecological cancer patients treated at the gynaecology clinic to participate in this study, with the study recruitment period being extended to six months. There were 120 patients who participated, reflecting a high participation rate (94%) in a public medical centre with among the highest oncology patient load nationally. The study centre (Universiti Malaya Medical Centre) is an academic tertiary-care healthcare centre that accepts referrals from healthcare centres across the country with a high patient load of more than a million clinical patient episodes per year. We would recommend future studies of gynaecological cancer in Malaysia to be conducted in multiple healthcare institutions to increase the number of eligible patients for recruitment. Conducting studies in multiple healthcare institutions in various states in Malaysia can also increase the generalisability of the findings to gynaecological cancer patients nationally.

It was observed that the sociodemographic characteristics of the study population were similar to the national population in the Malaysian National Cancer Registry Report [3], which indicated that it is likely to be representative of the general population of Malaysia. Many factors could have contributed to the small number of patients in this study, including issues with high attrition rates from clinical care, with anecdotal evidence of around 30% of cancer patients not remaining in clinical care. The relatively low survival rates of gynaecological cancers may also have led to fewer patients being available for interviews compared to the incidence rate of these cancers [4]. Similar studies conducted in Malaysia have had relatively small number of patients too, ranging from 115 to 138 study participants [11,44].

Additionally, future OOP expenditure studies can improve on the design of this study by conducting a prospective longitudinal study, which can thoroughly capture OOP expenditure without reliance on patient recall [45]. Due to resource and time constraints, this study applied a cross-sectional study design that relied on retrospective recall and patient-reported OOP expenditure [8,11,22]. Finally, it is possible that the findings of OOP expenditure in this study were underestimated or overestimated due to the retrospective design and recall bias [8,25], which given the relatively small sample size could have impacted the results. It is likely that the true financial burden of gynaecological cancer on patients and their households could be even higher than these findings. Additionally, since data collection for this study was conducted at the end of 2019, these findings did not reflect the impact of the global COVID pandemic on household income and impoverishment, which is expected to further influence OOP expenditure and CHE [46].

## 5. Conclusions

This study highlighted the high financial burden of gynaecological cancer patients in an upper-middle-income country, despite the availability of subsidised healthcare and health-related financial assistance in the study setting. While the quantitative value of the OOP expenditure was low compared to international currencies, the impact was substantial, with two thirds of the study households having experienced CHE. The majority of low-income households experienced CHE, which emphasises the need to assess the impact of health policies at the ground level in general and at the patient level in particular. These findings indicate the need to broaden the coverage of the existing government subsidy and health-related financial assistance provided for the medical needs of low- and middle-income households, and particularly the need to extend the coverage of current schemes to all adult and geriatric patients and for the healthcare subsidy to be more comprehensive for treatment received in all public hospitals. It is hoped that these findings can inform future policies to optimise equitable healthcare and eliminate health disparities in Malaysia.

## Figures and Tables

**Table 1 healthcare-10-02099-t001:** Sociodemographic and clinical characteristics of the study population (*N* = 120).

Variable	*n* (%)	Mean (SD)/Median (IQR)
Sociodemographic profile Age (years) Under 40 41–59 60 and older Ethnicity Chinese Malay Indian Other races	14 (11.7) 39 (32.5) 67 (55.8) 61 (50.8) 40 (33.3) 17 (14.2) 2 (1.7)	60 (14)/ 61 (18) (min = 22; max = 88) -
Education level No formal primary Primary Secondary Diploma Undergraduate degree Post-graduate degree	8 (6.7) 27 (22.5) 54 (45.0) 17 (14.2) 10 (8.3) 4 (3.3)	-
Healthcare insurance Yes No	9 (7.5) 111 (92.5)	-
Employment status Working full-time Working part-time Retired Not working	20 (16.7) 4 (3.3) 15 (12.5) 81 (67.5)	-
Household income (per month) Q1 Q2 Q3 Q4 Q5	24 (20.0) 24 (20.0) 24 (20.0) 24 (20.0) 24 (20.0)	809 (333)/980 (500) (min = 200; max = 1400) 1995 (443)/2000 (300) (min = 1450; max = 3250) 3000 (437)/3000 (835) (min = 2500; max = 3850) 5248 (871)/5000 (1500) (min = 4000; max = 6800) 14,104 (11,988)/10,000 (4000) (min = 7000; max = 60,000)
Variable		*n* (%)
Type of cancer Ovarian cancer Cervical cancer Endometrial cancer		52 (43.3) 37 (30.8) 31 (25.8)
Most recent staging of cancer Stage I Stage II Stage III Stage IV		47 (39.2) 15 (12.5) 22 (18.3) 36 (30.0)
Number of years since initial gynaecological cancer diagnosis 1st year 2nd year–5th year 6th year–10th year 11th year and above		17 (14.2) 49 (40.8) 24 (20.0) 30 (25.0)
Experienced disease progression in the past 12 months Yes No		10 (8.3) 110 (91.7)
Clinical treatment received in the past 12 months Cancer treatment with management of comorbidities Cancer treatment without management of comorbidities Cancer surveillance with management of comorbidities Cancer surveillance without management of comorbidities		26 (21.7) 19 (15.8) 69 (57.5) 6 (5.0)
Cancer * treatment received in the past 12 months (*n* = 45) Chemotherapy Radiotherapy Surgery Hormonal therapy Both chemotherapy and radiotherapy Both chemotherapy and hormonal therapy Both chemotherapy and surgery Both radiotherapy and surgery Chemotherapy, radiotherapy and surgery		10 (22.2) 1 (2.2) 5 (11.1) 2 (4.4) 2 (4.4) 1 (2.2) 17 (37.8) 2 (4.4) 5 (11.1)

* These percentages sum up to a total of 100%.

**Table 2 healthcare-10-02099-t002:** Components of total out-of-pocket expenditure incurred by households with gynaecological cancer (*n* = 120).

Components of Out-of-Pocket (OOP) Expenditure	Overall Results	
	%	Mean (RM) SD/Median (IQR) Mean (USD) (SD)/Median (IQR)
Medical charges and fees	30.4	RM 3710 (RM 11,130)/RM 218 (2286) USD 844 (USD 2532)/USD 50 (520)
Complementary and alternative therapy	14.1	RM 1727 (RM 7247)/RM 150 (780) USD 393 (USD 1649)/USD 34 (177)
Vitamins and supplements	12.6	RM 1544 (RM 2568)/RM 450 (2000) USD 351 (USD 584)/USD 102 (455)
Transport to study institution	9.7	RM 1186 (RM 1827)/RM 515 (1010) USD 270 (USD 416)/USD 117 (230)
Community pharmacy expenditure	6.6	RM 809 (RM 2393)/RM 58 (588) USD 184 (USD 544)/USD 13 (134)
Homecare expenditure	6.5	RM 797 (RM 2104)/RM 0 (38) USD 181 (USD 479)/USD 0 (9)
Laboratory and diagnostic investigations	6.0	RM 728 (RM 1624)/RM 3 (745) USD 166 (USD 370)/USD 1 (170)
Medication	5.5	RM 671 (RM 3746)/RM 0 (91) USD 153 (USD 852)/USD 0 (21)
Health treatment	2.8	RM 338 (RM 2502)/RM 0 (0) USD 77 (USD 569)/USD 0 (0)
Food expenditure	2.3	RM 278 (RM 557)/RM 60 (219) USD 63 (USD 127)/USD 14 (50)
Parking charges	1.0	RM 127 (RM 295)/RM 20 (126) USD 29 (USD 67)/USD 5 (29)
Accommodation expenditure	0.9	RM 111 (RM 774)/RM 0 (0) USD 25 (USD 176)/USD 0 (0)
Medical devices expenditure	0.9	RM 109 (RM 349)/RM 0 (0) USD 25 (USD 79)/USD 0 (0)
Nursing home expenditure	0.5	RM 63 (RM 685)/RM 0 (0) USD 14 (USD 156)/USD 0 (0)
Other expenditures related to health needs	0.2	RM 19 (RM 109)/RM 0 (0) USD 4 (USD 25)/USD 0 (0)
Total OOP expenditure per patient household	100	RM 12,218 (RM 17,255)/RM 6134 (13,243) USD 2780 (USD 3926)/USD 1396 (3013)

**Table 3 healthcare-10-02099-t003:** Total health-related out-of-pocket expenditure and proportion of households experiencing catastrophic health expenditure and living below the poverty line according to income quintiles (*n* = 120).

Household Income Quintiles	OOP Expenditure Mean (RM) (SD)/Median (IQR) Mean (USD) (SD)/Median (IQR)	Households That Experienced Catastrophic Health Expenditure (CHE) *n* (%)	Households Living below the National Poverty Threshold before OOP Expenditure *n* (%)	Households Living below the National Poverty Threshold after OOP Expenditure *n* (%)
Q1 (*n* = 24)	RM 3859 (RM 3962)/RM 2196 (6013) USD 878 (USD 901)/USD 500 (1368)	20 (83.3)	24 (100.0)	24 (100.0)
Q2 (*n* = 24)	RM 8698 (RM 7733)/RM 6798 (13,313) USD 1979 (USD 1759)/USD 1547 (3029)	19 (79.2)	21 (87.5)	23 (95.8)
Q3 (*n* = 24)	RM 11,261 (RM 12,226)/RM 6748 (13,247) USD 2562 (USD 2782)/USD 1535 (3014)	15 (62.5)	0 (0.0)	11 (45.8)
Q4 (*n* = 24)	RM 19,005 (RM 22,252)/RM 7422 (26,924) USD 4324 (USD 5063)/USD 1689 (6126)	15 (62.5)	0 (0.0)	5 (20.8)
Q5 (*n* = 24)	RM 18,267 (RM 25,442)/RM 11,207 (17,695) USD 4156 (USD 5789)/USD 2550 (4026)	8 (33.3)	0 (0.0)	2 (8.3)

**Table 4 healthcare-10-02099-t004:** Total health-related out-of-pocket expenditure and proportion of households experiencing catastrophic health expenditure and living below the poverty line according to household income categories (*n* = 120).

Household Income Categories	OOP Expenditure Mean (RM) (SD)/Median (IQR) Mean (USD) (SD)/Median (IQR)	Households That Experienced CHE *n* (%)	Households Living below the National Poverty Threshold before OOP Expenditure *n* (%)	Households Living below the National Poverty Threshold after OOP Expenditure *n* (%)
B40 (*n* = 56)	RM 6408 (RM 7022)/RM 3733 (7497) USD 1458 (USD 1598)/USD 849 (1706)	43 (76.8)	45 (80.4)	50 (89.3)
M40 (*n* = 58)	RM 17,976 (RM 22,409)/RM 8993 (22,397) USD 4090 (USD 5099)/USD 2046 (5096)	33 (56.9)	0 (0.0)	15 (25.9)
T20 (*n* = 6)	RM 10,781 (RM 7350)/RM 12,621 (15,730) USD 2453 (USD 1672)/USD 2872 (3579)	1 (16.7)	0 (0.0)	0 (0.0)

**Table 5 healthcare-10-02099-t005:** Estimated association between variables and catastrophic health expenditure using univariate and multivariate regression analysis according to income quintiles (*N* = 120).

Variables	Univariate Crude Odds Ratio (95% Confidence Interval)	*p*-Value	Multivariate Adjusted Odds Ratio (95% Confidence Interval)	*p*-Value
Monthly household income quintiles Q1 Q2 Q3 Q4 Q5	Reference 1.750 (0.442–6.928) 2.500 (0.613–10.195) 3.000 (0.774–11.627) 10.000 (2.545–39.293)	0.425 0.201 0.112 0.001	Reference 0.600 (0.146–2.473) 0.400 (0.102–1.572) 0.333 (0.086–1.292) 0.100 (0.025–0.393)	0.480 0.189 0.112 0.001 *
Stage of cancer at the beginning of the 12 months’ recall period				
Stage I and stage II	Reference		Reference	
Stage III and stage IV	2.762 (1.264–6.033)	0.011	2.049 (0.785–5.344)	0.143
Cancer treatment received during 12 months’ recall period				
Chemotherapy				
Yes	Reference		Reference	
No	0.206 (0.073–0.583)	0.003	0.120 (0.033–0.443)	0.001 *

* statistically significant at *p* < 0.05.

## Data Availability

Not applicable.

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
