# Peer review of "The Burden of Out-of-Pocket Expenditure Related to Gynaecological Cancer in Malaysia"

_healthcare, 2022, doi:10.3390/healthcare10102099_

Round 1
Reviewer 1 Report
The articles explored the OOP expenditure and CHE among patients with the three most common gynaecological cancers as well as the associated factors with CHE. In general, given such a limited sample size, the design and results are, to some extent, informative to the current issue in Malaysia. However, this study needs major improvement in its rationale, variables selection and table formatting.
1. Introduction
After reviewing the introduction, it is still unclear why the author picked up patients with gynaecological cancer as the focus of this study. As the author mentioned in line 36 “Treatment for patients with cancer generally involves high-cost clinical 36 strategies and medications”. That is, why do people with gynaecological cancer deserve special attention from the author, compared to patients with other types of cancers? This needs to be further explained in the introduction.
Also, it is better to have a paragraph in the introduction that helps frame the predictor variables used in the current study. Why did the authors choose Chemotherapy treatment as one of the predictors? What are the relative studies regarding those predictors?
The introduction also lacks the research hypothesis based on former studies and literature reviews.
2. Materials and Methods
In Line 85, the author needs to explain the rationale for why patients with multiple cancer diagnoses were excluded from the participant selection, as how this is aligned with the purpose of the study.
Also, I am still doubting how accurate these self-reported data on OOP expenditure could reach, even though the author already put limitation points. Given such a small sample size, even several inaccurate reports could have a major impact on the analysis results. This could be further discussed in the results or discussion.
In Line 118, “The association between independent factors and outcomes was assessed using binary logistic regression analysis”. What are the independent factors used here? please clearly stated them in this section
3 Results.
For Line 130-132, this information about patients could be moved to the data collection section
In the subsection titled “Health-related Out-of-pocket Expenditure”, what is the underlying logic for classifying participants based on both “Income quintile” and “ household income” variables? What is the benefit of categorizing patients in both ways for the current study?
The tables should be put in the bottom part of the manuscript, not in the middle. Also, the table looks messy, it is better if one table just appear on one page.
In line 169, what are the value for The Nagelkerke R2 and Hosmer-Lemeshow tests?
Author Response
Response to Reviewer 1 Comments
We thank the reviewers for these constructive comments. We have made revisions according to their suggestions, which we hope has improved the clarity of our manuscript.
Point 1: Introduction
After reviewing the introduction, it is still unclear why the author picked up patients with gynaecological cancer as the focus of this study. As the author mentioned in line 36 “Treatment for patients with cancer generally involves high-cost clinical 36 strategies and medications”. That is, why do people with gynaecological cancer deserve special attention from the author, compared to patients with other types of cancers? This needs to be further explained in the introduction.
Response 1: Further explanation on why gynaecological cancers were the focus of this study has been added to the manuscript [Page 1 (Lines 34-42)]. Gynaecological cancers are among the top cancers in female patients in Malaysia with high mortality, with scarce data and few studies conducted on the healthcare and economic burden of this patient population. This study aimed to describe the economic burden of this disease to patients and their households in terms of OOP expenditure and CHE. We have also added text to the Discussion section [Page 7 (Lines 317-321)] citing a study in Malaysia that found cancer to be the only comorbidity that influenced CHE among the elderly.
Point 2: Also, it is better to have a paragraph in the introduction that helps frame the predictor variables used in the current study. Why did the authors choose Chemotherapy treatment as one of the predictors? What are the relative studies regarding those predictors? The introduction also lacks the research hypothesis based on former studies and literature reviews.
Response 2: A paragraph on predictor variables had been added in the introduction [Page 2 (Lines 86-97)]. Related studies regarding those predictors had been added as well [Page 2 (Lines 71-74) and Page 2 (Lines 86-97)]. References to published studies and related literature reviews had been cited and added to the manuscript.
Point 3: Materials and Methods
In Line 85, the author needs to explain the rationale for why patients with multiple cancer diagnoses were excluded from the participant selection, as how this is aligned with the purpose of the study.
Response 3: The explanation of the rationale of inclusion and exclusion criterias was added [Page 3 (Lines 112-120)].
Point 4: Also, I am still doubting how accurate these self-reported data on OOP expenditure could reach, even though the author already put limitation points. Given such a small sample size, even several inaccurate reports could have a major impact on the analysis results. This could be further discussed in the results or discussion.
Response 4: Sample size justifications were added in the Materials and Methods section [Page 3 (Lines 130-137)] and the Discussion section [Page 7-8 (Lines 353-371) and Page 8 (Lines 375-381)].
Text has also been added to the Discussion section [Page 8 (Lines 386-389)] to acknowledge that “given the relatively small sample size, retrospective design and recall bias could have impacted the results of this study”.
Point 5: In Line 118, “The association between independent factors and outcomes was assessed using binary logistic regression analysis”. What are the independent factors used here? please clearly stated them in this section
Response 5: The independent factors are now stated in the Materials and Methods section [Page 4 (Lines 159-160)] and the Results section [Page 5 (Lines 220-222)].
Point 6: Results.
For Line 130-132, this information about patients could be moved to the data collection section
Response 6: Line 130-132 had been moved to the data collection section as suggested by reviewer [Page 3 (Lines 134-137)].
Point 7: In the subsection titled “Health-related Out-of-pocket Expenditure”, what is the underlying logic for classifying participants based on both “Income quintile” and “ household income” variables? What is the benefit of categorizing patients in both ways for the current study?
Response 7: Participants were classified as “Income quintiles” to facilitate comparisons with published studies internationally. “Household income” was included as well because in Malaysia, income is classified as B40, M40 and T20 in the national statistic data, as referred to that defined in Department of Statistics Malaysia in year 2020, stated in the Introduction section [Page 2 (Lines 51-56)] and the Results section [Page 5 (Lines 201-206) and Page 5 (Lines 227-243)]. Text has also been added to Materials and Methods section [Page 4 (Lines 170-174)] with details on how the results are presented in the manuscript.
Point 8: The tables should be put in the bottom part of the manuscript, not in the middle. Also, the table looks messy, it is better if one table just appear on one page.
Response 8: As for the recommended formating, all tables have been moved to the bottom part of the manuscript and it has been formatted so that there is only one table on one page [Page 10-16].
Point 9: In line 169, what are the value for The Nagelkerke R2 and Hosmer-Lemeshow tests?
Response 9: The value for The Nagelkerke R2 and Hosmer-Lemeshow tests had been added in the Results section as suggested by the reviewer [Page 5 (Lines 225-227)].
Reviewer 2 Report
You should regress CHE on the sociodemographic factors as it will improve the research.

Author Response
Response to Reviewer 2 Comments
We thank the reviewers for these constructive comments. We have made revisions according to their suggestions, which we hope has improved the clarity of our manuscript.
Point 1: Abstract: The abstract is well-written, and it includes all crucial information. There is existing healthcare-related financial assistance for Malaysians from low- and middle-income households (i.e., MySalam and Peka B40 schemes) and those that seek care from government health facilities. The results depict that highest income quintile households (Q5) were less likely to incur CHE. Even though you are recommending the coverage of existing financial assistance to be broadened, could you be a little specific or expatiate further? What specific coverage should be broadened as per this study? You made a very good case in the “Discussion section” that the MySalam scheme should be extended to include geriatric patients of all ages. Could you bring that in the “Abstract section as well?” Also, if those from low-income households are more likely to incur CHE, then the Peka B40 scheme could also be increased to cover other non-medical expenditures using vouchers and the like.
Response 1: The recommendations to broaden the coverage of existing financial assistance have been added to the Abstract section [Page 1 (Lines 25-28)].
The recommendations on broadening the coverage of existing financial assistance have been expanded in the Discussion section [Page 7 (Lines 311-321) and Page 7 (Lines 322-337)].
Point 2: Introduction: The introduction is very well-written. However, you could provide more information on the MySalam and Peka 40 schemes. Why and when were they implemented? How impactful have they been? How will the results of this research be important to these existing health policies? Check line 41 (i.e., patients’) and correct it. Did you mean “financial” risk of cancer in line 42?
Response 2: Additional information on the MySalam and Peka B40 schemes have been added in the Introduction section [Page 2 (Lines 56-74)] and the Discussion section [Page 7 (Lines 311-321) and Page 7 (Lines 322-329) and Page 7 (Lines 334-340)] as recommended by the reviewer.
Line 41 has been amended as suggested by the reviewer. [Page 2 (Line 47)]
The sentence in Line 42 has been removed.
Point 3: Materials and Methods: When did the data collection begin and when did it end? Why did you choose only 120 participants?
Response 3: The data collection period has been added as recommended [Page 3 (Line 129)].
The reason of 120 participants included in the study had been included in the Materials and Methods section [Page 3 (Lines 130-137)] and the Discussion section [Page 7-8 (Lines 353-371) and Page 8 (Lines 375-381)].
Point 4: Based on the literature, people have examined the socio-economic determinants of CHE. Is there any reason these variables are not seen in the regression models? How do we then know whether the regression results are spurious or robust? In essence, the results of the included variables could be insignificant if we control for socio-economic and demographic variables. You should also include the regression equation that informed the regression analysis (such as Y=Bo + BiXi) and the CHE formula (if possible). You should explain the regression variables further in terms of the dependent and independent variables. I think the income quintile may be the independent variable and the CHE is the dependent variable. You should also explain further as to how each variable was measured and how, especially, the measurement of the dependent variable affected your choice of regression model.
Response 4: Independent variables and dependent variables had been used as recommended by reviewer [Page 4 (Lines 159-160)].
All of the independent variables were assessed in bivariate analysis. Other variables were not significantly associated with CHE in bivariate analysis and thus not included in the univariate regression analysis. Further explanation had been added as suggested by reviewer, reported in the Materials and Methods section [Page 4 (Lines 159-166)] and the Results section [Page 5 (Lines 220-224)].
The general definition for CHE is included in the Introduction section [Page 2 (Lines 79-85)] and the specific definition for CHE that was used in this study was stated in the Materials and Methods section (Data collection subsection) [Page 4 (Lines 148-149)].
Point 5: Results: The participation rate was incredible. Table 1 summarizes the socio-demographic and clinical characteristics, and they should be added to the multivariate regression analysis as control variables. For instance, the distance to healthcare facility may have impact on CHE and so could health insurance subscription status.
Response 5: Household income, cancer stage and receiving chemotherapy were the variables included in the multivariate regression model [stated in Page 5 (Lines 220-222)]. Other variables were not significantly associated with CHE in bivariate analysis and thus not included in the univariate regression analysis, reported in the Materials and Methods section [Page 4 (Lines 159-166)] and the Results section [Page 5 (Lines 220-224)] and in [Table 5 and 6].
Point 6: Check this paper “socio-demographic, cognitive status and comorbidity determinants of catastrophic health expenditure among elderly in Malaysia” published in the International Journal of Economics and Management by Koris et al. (2017).
Response 6: Thank you for bringing Koris et al. 2017 to our attention. We have added a reference to this study in the Discussion section [Page 7 (Lines 317-321).
Point 7: Discussion: This section was very well expounded. The study gave specific recommendations to include geriatric patients of all ages in MySalam. It also discussed the unknown reasons why people choose to attend hospitals that are not covered by Peka B40 for healthcare services. Probably, future research could look into it. You justify the small sample size based on the fact that the socio-demographic characteristics of the study population was similar to the national population. However, you did not regress the CHE on the socio-demographic characteristics.
Response 7: All of the variables in Table 1 including socio-demographic characteristics were included in bivariate analysis, stated in the Materials and Methods section [Page 4 (Lines 159-160)]. Household income, cancer stage and receiving chemotherapy were the variables included in univariate and multivariate regression model, stated in the Materials and Methods section [Page 4 (Lines 160-166)] and the Results section [Page 5 (Lines 220-224)] and in [Table 5 and 6]. Other variables were not significantly associated with CHE in bivariate analysis and thus not included in the univariate regression analysis, reported in the Materials and Methods section [Page 4 (Lines 159-166) and the Results section [Page 5 (Lines 220-224)] and in [Table 5 and 6].
Round 2
Reviewer 1 Report
I think this revision looks much better!
Author Response
We thank the reviewers for this comment.
Reviewer 2 Report
It is a well-written article.
Author Response

(The authors gave the same response as above.)
